# Programme and policy perspectives towards a tobacco-free generation in India: findings from a qualitative study

Shalini Bassi [1,2] Monika Arora [1,2] Nishibha Thapliyal,[1] Muralidhar M Kulkarni,[3] Rohith Bhagawath,[3] Ilze Bogdanovica,[4] Veena G Kamath [3] John Britton,[5] Manpreet Bains [4]

¹Health Promotion Division, Public Health Foundation of India, Gurugram, India
²HRIDAY, New Delhi, India
³Community Medicine, Kasturba Medical College, Manipal Acadamy of Higher Education, Manipal, Karnataka, India
⁴School of Medicine, University of Nottingham, Nottingham, UK
⁵Epidemiology and Public Health, University of Nottingham, Nottingham, UK

**Correspondence to**
Shalini Bassi;
shalini.bassi@phfi.org

## ABSTRACT

**Objective** This study explored multistakeholder perspectives on existing adolescent-specific tobacco control policies and programmes, to advance India's transition towards a tobacco-free generation.
**Design** Qualitative semi-structured interviews.
**Setting** Interviews were conducted with officials involved in tobacco control at the national (India), state (Karnataka), district (Udupi) and village level. Interviews were audio recorded, transcribed verbatim and analysed thematically.
**Participants** Thirty-eight individuals representing national (n=9), state (n=9), district (n=14) and village (n=6) levels, participated.
**Results** The study findings highlighted the need to strengthen and amend the existing Tobacco Control Law (2003) provisions, particularly in the vicinity of schools (Sections 6a and 6b). Increasing the minimum legal age to buy tobacco from 18 to 21 years, developing an 'application' for 'compliance and monitoring indicators' in Tobacco-Free Educational Institution guidelines were proposed. Policies to address smokeless tobacco use, stricter enforcement including regular monitoring of existing programmes, and robust evaluation of policies was underscored. Engaging adolescents to co-create interventions was advocated, along with integrating national tobacco control programmes into existing school and adolescent health programmes, using both an intersectoral and whole-societal approach to prevent tobacco use, were recommended. Finally, stakeholders mentioned that when drafting and implementing a comprehensive national tobacco control policy, there is a need to adopt a vision striving toward a tobacco-free generation.
**Conclusion** Strengthening and developing tobacco control programmes and policies are warranted which are monitored and evaluated rigorously, and where adolescents should be involved, accordingly.

## BACKGROUND

Tobacco use is a leading risk factor associated with several diseases, premature death and disability. It is a major threat to developing and low-income/middle-income countries such as India, where tobacco use results in 1.35 million deaths, annually.[1 2] Furthermore, of all deaths related to non-communicable

## STRENGTHS AND LIMITATIONS OF THIS STUDY

⇒ This study captures perspectives on existing tobacco control policies in India, from a diverse range of stakeholders representing different levels of governance (national/state/district/village).
⇒ Perspectives were captured on existing tobacco control policies and programmes, barriers and facilitators to their successful implementation and suggested recommendations for strengthening existing and developing new policies to achieve the ultimate goal of a tobacco-free generation for India as well as replicate such models globally.
⇒ Despite the fact that the interviewees for the in-depth interviews were not randomly selected, our study findings are still significant and transferable.

diseases in India, in 2019, approximately 1.08 million deaths (17.7%) were due to tobacco use.[3] The prevalence of tobacco use among adults in India has declined from 34% to 28% between 2010 (Global Adult Tobacco Survey (GATS-1)[4] and 2017 (GATS-2)[5] and among school-going students aged 13–15 years from 16.9% (Global Youth Tobacco Survey (GYTS-1, 2003)[6] to 8.5% (GYTS-4, 2019).[7] The majority of adult tobacco users initiate use during adolescence,[8] often leading to lifelong use, with low quit rates.[9] Therefore, prevention of tobacco uptake among children and adolescents is crucial to reducing the harms caused by tobacco use. There are many tobacco control programmes and policies, but the need for age-specific policies has gathered prominence in recent decades. Adolescent-specific policies are warranted, especially, preventing uptake in the first place.[10]

The Government of India (GoI) has implemented various policies and programmes to prevent tobacco uptake among children and adolescents. Section 6a of the Cigarettes and Other Tobacco Products (Prohibition of Advertisement and Regulation of Trade

and Commerce, Production, Supply and Distribution) Act (COTPA) 2003,[11] stipulates the prohibition of the sale of tobacco products to and by minors; and Section 6b, prohibits the sale of tobacco products within the surrounding 100 yards of educational institutions.[11] As part of the 'National Tobacco Control Programme' (NTCP), India also launched the School Programme across public and private schools, to equip young people with the knowledge and appropriate skills to make informed decisions related to tobacco consumption and build understanding on the consequences of tobacco use.[12] To reinforce such efforts, the Ministry of Health and Family Welfare (MoHFW)-GoI also introduced 'Tobacco-Free Educational Institution' (ToFEI) guidelines,[13] for all education establishments, across India, to generate awareness about the harmful and long-term effects of tobacco use and to create healthy tobacco-free educational institutes.

Despite these efforts, the implementation of tobacco control policies continues to be a challenge.[14] The evidence is mixed on whether investing in youth and adolescent tobacco control and prevention efforts is effective.[15] A growing focus is on 'anti-tobacco' interest groups and their influence on policy implementation and success.[16] Multiple partnerships have been found to be beneficial in tobacco control policy implementation at various levels.[17 18] Thus, considering the perspectives of various stakeholders for formulating tobacco control policies might be the way ahead. Perspectives from multiple stakeholders increases the range and quantity of information for researchers and thus increases the likelihood of actions being developed and implemented.[19]

This study explored the views of a diverse range of stakeholders (including policymakers, policy implementers, representatives from civil society, implementation agencies and other stakeholders) on India's existing tobacco control policies and programmes focusing on adolescents, barriers to and facilitators for their successful implementation and suggested recommendations for the future.

## METHODS
### Design
Qualitative, one-to-one, semi-structured interviews were conducted with key officials at the national (India), state (Karnataka), district (Udupi) and village level.

### Patient and public involvement
Patients or the public were not involved in the design, conduct, reporting or dissemination plans of our research.

### Recruitment and sampling
A purposive sampling strategy was developed[19 20] where a list of 51 eligible individuals was drafted by members of the research team (MA, SB and MMK), based on professional networks and level of engagement with tobacco control programmes, at different levels of governance. The interviewees varied in terms of their roles, responsibilities and seniority, to ensure that a wide range of perspectives were captured. Potential interviewees were approached by SB and RB via email and telephone. A study information sheet and consent form were sent prior to completing interviews, and an interview was arranged at a time convenient for each individual.

### Data collection and procedure
A semi-structured interview guide was developed to explore interviewees' views on existing adolescent-specific tobacco control policies and programmes in India, barriers and facilitators to successful implementation, as well as recommendations for their successful implementation and strengthening, to advance India's transition towards a tobacco-free generation. The interview guide was developed by SB, MB, with inputs from MA, MMK and VG, and was based on the School Programme component delivered as part of NTCP,[12] ToFEI guidelines by MoHFW,[13] provisions of the Indian Tobacco Control Act (2003),[11] that specifically addressed the prevention of tobacco uptake and use among children and adolescents (online supplemental table 1).

Interviews were conducted between July 2019 and September 2021, in English by trained qualitative researchers. National and state-level stakeholder interviews were moderated by SB, while district and village level by MMK and RB. Due to the COVID-19 mitigation measures in India (eg, travel restrictions), national and state-level interviews were conducted virtually using Zoom (during December 2020 to September 2021), while district and village level interviews were conducted in-person (from July 2019 to March 2020). The interviewees were asked to review and sign a consent form prior to interview (verbal consent was sought for virtual interviews). The consent process covered permission for data collection, audio recording of interviews, data to be used in research publications and dissemination while maintaining interviewee confidentiality and anonymity.

### Data analysis
Interviews were transcribed verbatim by members of the research team. Identifiable data were anonymised and double-checked for accuracy by senior members of the research team. Transcripts were analysed by SB, NT and MB using thematic analysis[21] where both inductive and deductive approaches[22] were adopted. To ensure the validity of interpretations,[23] each individual familiarised themselves with the data separately which led to the identification of initial codes. Further readings resulted in the development of more substantive themes and subthemes. Following this, a series of virtual calls were organised between SB, NT and MB to discuss preliminary themes which led to a more definitive thematic framework, which was agreed and then applied to the remaining data (by NT). NVivo 12.0 was used to facilitate the analysis process and subsequent synthesis and reporting.

**Table 1** Demographic characteristics of the study interviewees (N=38)

|  | N (%) |
|---|---|
| Sex | |
| Male | 26 (68.4) |
| Female | 12 (31.6) |
| Geographical representation | |
| National | 9 (23.7) |
| State | 9 (23.7) |
| District | 14 (36.8) |
| Village level | 6 (15.8) |
| Years of working experience in tobacco control | |
| 0–5 years | 9 (23.7) |
| 6–10 years | 7 (18.4) |
| >10 years | 10 (26.3) |
| Not available | 12 (31.6) |
| Type of organisation | |
| Government | 26 (68.4) |
| Civil society representatives | 11 (29) |
| Development agency | 1 (2.6) |

## RESULTS

Thirty-eight interviews were conducted with: 9 national-level, 9 state-level (Bengaluru, Karnataka), 14 district (Udupi) level and 6 village-level stakeholders (table 1). Interviewees included officials from the NTCP, members of the tobacco control committee (national, state and district level), representatives from civil society organisations working in tobacco control, health, education, agriculture, food and civil supplies, police, parliamentary affairs, district administration, rural development and Panchayati Raj (a system of local self-government of villages in rural India). Interviews averaged 44 min (range 20–111 min).

### Thematic analysis

Four themes were generated: (1) Views on existing tobacco control policies and programmes focusing on adolescents in India; (2) Barriers and facilitators to successful implementation; (3) Recommendations for successful implementation; and (4) Strengthening adolescent-specific tobacco control policies and programmes to advance India's transition towards a tobacco-free generation.

### Views on existing tobacco control policy and programmes focusing on adolescents in India

Interviewees reported many tobacco control programmes and policies to prevent tobacco uptake among adolescents. They believed school-based programmes or interventions to have an important role in preventing adolescents from using tobacco. Most of the interviewees spoke about the 'School Health Programme', which forms part of NTCP, and a few also mentioned smaller regional or local initiatives such as, the 'Yellow line campaign' (wherein, two yellow lines are painted 100 yards away from the boundary wall of an education institute demarking a 'No Tobacco Zone'); the 'Rose Campaign' (where a rose flower was given to petty shop owners selling tobacco and were made aware of its consequences and asked to display boards with tobacco warnings); the 'National Population Education Project' (creating awareness or helping adolescents understand the impact of using tobacco through several activities, such as poster competitions, memes, folk dance, skits, role play, mono acting) and also other community-based activities with the involvement of local leaders and the formation of a 'Taluka level coordination committee' (a group of several villages counted together for administrative purposes), for controlling tobacco in the community.

As part of the NTCP, the interviewees mentioned the ongoing sensitisation activities (sessions/talks/audio-visual clips) school children were exposed to, as well as the formation of anti-tobacco school committees, making an anti-tobacco pledge in school assembly, the organisation of anti-tobacco themed competitions like drawing, posters, painting, quiz competitions and relay-races. Some spoke of annual competitions to formulate an anti-tobacco message and role-play activities, along with cultural-day programmes in schools, to generate awareness of tobacco, among both students and parents. A representative from the State Education Department mentioned that many school teachers are also invited to be a part of government-organised sensitisation training, which aims to create awareness about the consequences of using tobacco for school children, where manuals were reportedly provided to school authorities, to aid the process.

> We have received a manual, wherein information about cigarette content, and activities to be taken by the students to not fall into the trap is provided. There are many pictures in that, wherein it is displayed about the effects of usage of tobacco, and many more things are there in the manual. There are several activities and content that create awareness among students and we have already passed this to school authorities.

Many interviewees also discussed the ToFEI guidelines developed by the MoHFW. A national-level non-governmental organisation (NGO) representative highlighted that the most important indicator in ToFEI, deals with the implementation of section 6b of COTPA around the prohibition of the sale of tobacco near educational institutes:

> One of the great outcomes that have come out of this ToFEI programme is section 6b of COTPA which prohibits any shop selling tobacco within 100 yards of any educational institute, right? And the industry makes sure that the shops are there. One of the components in these guidelines makes us report the number of tobacco shops which are around these schools.

## Barriers and facilitators to successful implementation of tobacco control programmes and policies

Most interviewees reported weak monitoring and enforcement of 'NTCP' and 'ToFEI guidelines' as the primary barrier to their successful implementation. A national level civil society representative stated that, 'there is no mechanism to monitor how many schools are tobacco-free in the state'. Another interviewee mentioned that the guidelines were limited in terms of their reach, 'The guidelines are not reaching upto where it has to, like if we talk about educational institutions. The Principals, the administrative authorities of these educational institutions seem to be unaware and they are unaware because nobody has reached out to them'. Some also referred to a lack of dedicated staff to carry out programme activities and budget as barriers to programme implementation. A national-level representative said, 'There are 75 districts, there is staff only in 26 districts, 26 districts do proper reporting and monitoring but the remaining districts are not able to do because there are no staff specific to our department'.

Some even mentioned the lack of subnational level representation during the policy development process as a barrier. Many stated a lack of awareness about the ongoing nature of these programmes, the Tobacco Control Act, and its provisions among members of enforcement agencies (organisations that are responsible for programme implementation education, police department, etc), were barriers to successful implementation of programmes. The COVID-19 pandemic was also mentioned by some as a barrier, because of the way it has overwhelmed many systems, especially within health and education, due to schools being closed for more than one academic year making tobacco control trivial.

Many interviewees proposed a bottom-up approach to facilitate proper implementation of existing programmes and policies. Some suggested prioritising the NTCP programme at all levels, especially subnationally, along with amalgamating school-based programmes under one umbrella, to facilitate better implementation of the NTCP. Some even mentioned implementing programmes (like NTCP and ToFEI guidelines) in coordination with other ministries, especially the police and education departments, for better coverage and enforcement. A national representative proposed a partnership between the Ministries of Health and Education to facilitate implementation of the ToFEI guidelines. Some representatives also mentioned that tobacco control needs to be incorporated and discussed in state and district-level developmental meetings. A state-level government representative said, 'District level Coordination Committee' meeting would be a very good platform where such issues can be highlighted and brought to notice of the deputy commissioner himself and when he pitches in.'

## Recommendations for successful implementation of tobacco control policies and programmes

### Awareness generation and capacity building of enforcement agencies

While talking about the lessons and recommendations for strengthening adolescent-specific tobacco control efforts, most interviewees underlined the need to continuously educate adolescents about the consequences of using tobacco and how the tobacco industry uses deceptive tactics to entice them to take up their products. Using explicit examples, such as the techniques used by the tobacco industry were highlighted. More use of IEC (Information, Education and Communication) material, videos, short films and awareness campaigns, were suggested, along with the use of social media platforms (*Facebook, Twitter and WhatsApp),* due to their popularity among the younger generation. Many stressed the need for targeted sensitisation programmes for enforcement agencies on the importance of enforcing tobacco control laws.

Interviewees believed school-based interventions to be an important way to prevent adolescents from using tobacco. A state representative said to 'Take up important aspects of tobacco education into the curriculum of students wherein they are taught actively that, these are habits that are detrimentally harmful, and the industry is going to try and fool them and to look for industry interference'. Some interviewees also suggested co-creating adolescent-driven projects or interventions[24] engaging them in creating awareness among their peers, where an example from a school-based intervention delivered in Delhi schools was referred to:

> The MYTRI trial, "if you see this element of peer activism in terms of interventions. So, if you do school-based interventions which are actively involving youth, as a change maker or as advocates, those interventions are more effective".

### Strengthening the enforcement and monitoring of existing tobacco control policies and programmes

A state civil society representative suggested the education department needed to develop a monitoring tool to ensure compliance with ToFEI guidelines.

> The educational department has to come up with a monitoring tool to assess the level of compliance with ToFEI. Practically someone from the education department has to physically verify the compliance. The school may upload the photographs but the physical verification by the concerned officers of the educational department and sharing the monthly progress report with the key stakeholders is more helpful to understand the level of compliance.

A national-level interviewee pointed out that the ToFEI guidelines need to be implemented properly, especially in village schools, where teachers must first be oriented, and better collaboration between ministries needs to be

addressed. In rural areas, integrating ToFEI guidelines in 'Aadarsh gaon Yojana' (Model village, Rural Development Model)[25] which focuses on village development and includes social, cultural development and social mobilisation of the village community; or Clean India Mission (Swachh Bharat Abhiyan)[26] a country-wide campaign initiated by the GoI to eliminate open defecation and improve solid waste management, was suggested.

To reduce and prevent tobacco use among youth, most interviewees emphasised establishing a comprehensive enforcement mechanism for implementing the COTPA Act, in the vicinity of educational institutions. Many even stated that specific amendments to COTPA-2003, especially surrounding educational institutes, such as increasing penalties/fines, were needed. The need for strengthening implementation of 6a and 6b of COTPA-2003 was highlighted by a state level civil society representative, 'So we have COTPA section 6a, and 6b provision. It says tobacco cannot be sold within 100 yards of any educational institution, so if you implement this particular section 100% in any state, any geographical area, then no chance of accessibility or availability of tobacco products'. Including section 6b in the 'regular checklist' for the school's inspector, along with strict enforcement in and around schools, was also advocated.

The significance of robust regular monitoring at every level was underscored by interviewees. All tobacco-related programmes, particularly those in schools, must be monitored by the government. The school's involvement in monitoring was considered critical. An official from the health department said 'we conduct a programme in one school and go back to that school after 4 years. So there need for proper monitoring of every school. So, there are 3000–4000 schools in every district. Covering all schools at such large scales can only be done through district human resources'. Similarly, many interviewees also suggested engaging civil societies, student self-help groups (like Bharat Scouts, National Cadet Corps), and police functionaries for monitoring and enforcement, to maximise effectiveness and increase programme accessibility.

### Multisectoral and multistakeholder engagement
All interviewees commented on the successful implementation of any programme being dependent on having effective and efficient intersectoral collaborations, from national to grassroot levels (Panchayati Raj Institutions at the village level). The active involvement of all stakeholders, including health, non-health and NGOs, to safeguard adolescents from tobacco was also highlighted, 'we need to wage a battle. Finance, human resources, NGOs, policy, intersectoral collaboration not just within the health sector but with other departments also'. Many interviewees suggested integrating the NTCP into existing health and adolescent health programmes like the Ayushman Bharat Scheme,[27] National Adolescent Health Programme (Rashtriya Kishor Swasthya Karyakram)[28] and, National Programme for Prevention and Control of Cancer, Diabetes, Cardiovascular Diseases and Stroke.[29]

We have a national tobacco control programme, we have a district cancer control programme, we have oral health programme, multiple tobacco control programmes which are being implemented. We have district tobacco control programme. There are multiple overlapping issues with each of these programmes, there has to be coordinated way and mechanism in which this could be implemented at the ground level and there has to be consistency in the way in which it has to be implemented.

### Regular impact evaluation
Interviewees underscored the importance of routine programme and policy evaluation, particularly when considering the evolving nature of tobacco industry tactics to target adolescents. Research studies could help in identifying the loopholes or lacunae of the ongoing programmes by providing evidence and encouraging them to take corrective actions to improve them. It was stated that there is a need to identify opportunities and weaknesses in the context of regional, state and district variations, and thus, a multipronged approach should be developed accordingly, 'We need to calibrate intervention, and policies as per the development of industry and science and we regularly need to undertake impact evaluation studies to understand what are the gaps and what we have achieved'.

### Strengthening adolescent-specific tobacco control policies and programmes to advance India's transition towards a tobacco-free generation
#### Development of new tobacco-free generation-centric policies
Many interviewees highlighted the need for new and achievable policies which focus on realising a tobacco-free generation. Considering the magnitude of smokeless tobacco use in India, many interviewees suggested, shift in focus of policymaking, from only smoking to both smoking and smokeless tobacco. Many even suggested increasing the minimum legal age to buy tobacco from 18 to 21 years. Vendor licensing was also identified as a key strategy to reduce accessibility to tobacco. A stakeholder from the state said, 'What happens when a child goes to buy chocolate or cakes and sees lots of tobacco products also…so there is a chance he may buy tobacco products also. Because everything is sold in one shop so there is a risk of a small child getting access to a tobacco shop. So here, by vendor licensing, what we are doing is, we are preventing accessibility of a child getting tobacco'.

Interviewees also spoke about developing policies or regulations to prohibit tobacco advertising, promotion and sale on 'Over The Top' (OTT) platforms. These are needed to protect minors from all direct and indirect methods (surrogate promotion by big tobacco) of promotion and sales on OTT platforms. E-cigarette sales on online platforms was referred to as an example of how compliance with the Prohibition of Electronic Cigarettes (Production, Manufacture, Import, Export, Transport, Sale, Distribution, Storage and Advertisement) Act,

 

2019 in the country is weak.[30] The interviewees stressed the need for strict monitoring of regulations on OTT platforms.

## DISCUSSION

The study emphasised the need for educating adolescents, communities, teachers and enforcement agencies on the consequences of tobacco use, tobacco control laws and the tobacco industry's deceptive tactics to target young people. For preventing tobacco use among adolescents, our study demonstrates the need to formulate policy around realising a tobacco-free generation, engaging adolescents meaningfully, co-creation of adolescent-specific tobacco control interventions, strengthening and amending COTPA provisions surrounding educational institutions, improving multisectoral and multistakeholder engagement, adopting the whole-societal approach, policies prioritising smokeless tobacco control and future research for identifying the loopholes or lacunae of the ongoing programmes. The importance of having robust monitoring and evaluation mechanisms was underscored, to ensure the most effective interventions and policies are in place.

The introduction of COTPA[11] and ToFEI guidelines[13] were considered prime steps towards tobacco control among adolescents. But even after years since these have been implemented, awareness and compliance remain a major concern, especially in rural settings and among enforcement agencies.[31–35] The study highlights a need to increase awareness among adolescents, teachers, community members and law enforcement agencies, which is supported by previous research.[36–38] Evidence from Maharashtra (India) showcased the positive effects of sensitisation efforts with police on enforcement of COTPA with an increased collection of 'challans' (fines)[39] and action against shops violating section 6b of COTPA.[38]

The need to develop adolescent-driven programmes to meaningfully engage adolescents, wherein they serve as ambassadors and be involved in creating awareness among their peers, their families and the community about tobacco use, was repeatedly emphasised in our findings. Many successful youth driven tobacco-based initiatives[40–42] have observed favourable results in terms of policy development,[43] sensitisation and reduced current tobacco use among youth.[44] A campaign engaging National Service Scheme (NSS) volunteers from 540 colleges in Mumbai resulted in improving awareness of tobacco control issues among 176 000 students,[45] and the use of NSS volunteers[46] was also suggested in our study to support tobacco control initiatives.

This study underscored the importance of using social media to promote tobacco control messages and restricting tobacco industry marketing on social media, due to its widespread use among youth.[47] Existing evidence supports the use of social media for tobacco control.[48–52] For example, a recent social media campaign sought to educate high school students on e-cigarettes led to greater knowledge and beliefs about the harmful effects of e-cigarette use, suggesting social media as a promising tool for tobacco education, among young people.[53] Another study also demonstrated the use of digital media to empower adolescents in smoking prevention.[54] The need for extending the Tobacco Advertising, Promotion and Sponsorship (TAPS) ban to cover OTT platforms was underscored in our study. There is ample evidence worldwide showcasing the use (direct and indirect) of social networking sites for TAPS specifically targeting youth.[55–58] By using social media influencers, tobacco giants have found a way to circumvent policy guidelines.[59] Singapore passed laws on cyber surveillance and social media monitoring to counter the tobacco menace[60] and has also penalised and taken legal action on the TAPS violation.[61] A comprehensive ban on TAPS to ensure online streaming platforms are compliant with COTPA, is crucial for India to protect children from exposure to tobacco products displayed on the OTT platforms.Our study also recognises the need to amend existing policies and development of new policies from tobacco control and prevention to 'Tobacco-Free future Generation' or 'Tobacco Endgame', to make the next generation 'tobacco-free'. Several countries like Scotland,[62] Singapore[63] New Zealand,[64] Finland[65] and others are striving towards a tobacco-free goal. A study weighing up the impact of New Zealand's Tobacco-Free Generation policy found that the implementation of Tobacco Free Generation results in a substantial decrease in smoking prevalence.[66] India also needs to undertake strategic steps towards a tobacco-free generation by coordinating ongoing efforts of implementing ToFEI guidelines and global youth campaigns like NMT21C (No More Tobacco in the 21st Century).[67]

India should also consider making policy changes in the minimum legal age to buy tobacco, raising it from 18 to 21 years, as suggested by the stakeholders in our study. The evidence shows a high percentage of smokeless (75%) and smoking (77%) tobacco users in India initiate use before or up to the age of 21 years.[68] This is also supported by the scientific evidence that the brain continues to develop until about the age of 25 years. During this time period, brain growth is not complete and susceptibility to the damaging effects of tobacco smoke may be enhanced.[69 70] The probability that an adolescent will remain tobacco-free for the rest of his/her life is higher. The effectiveness of this approach can be seen in California where 'Tobacco 21' was passed in 2016 and this law has led to a decrease in the illegal sale of tobacco[71 72] and a decline in cigarette smoking in high school students.[73]

India also faces the dual challenge of dealing with tobacco smoking and smokeless tobacco, especially in rural areas where the prevalence of smokeless tobacco use is much higher.[74] Additionally most smokeless tobacco users are dual users[75] (who use both smokeless and smoking tobacco). Furthermore, current tobacco laws in the country are oriented towards smoking tobacco.

Hence, the need for the development of new policies to cater for smokeless tobacco users was also emphasised in both our and previous studies.[76] Vendor licensing as recommended in our study can help in reducing the vendor density, especially near schools,[77] and lower the odds of cigarette use initiation in adolescents.[78]

Findings from our study prioritises a need to strengthen the school health component which forms part of the ongoing NTCP, as school settings are deemed best for tobacco use prevention and addiction for children, school staff and families. Families have an enormous influence on a child's tobacco use behaviour[79 80] and this has also been reported in our study. Thus, parents, particularly in villages, need to be sensitised and should be involved in school-based tobacco-free efforts, to reduce tobacco exposure at home.

In terms of stricter implementation, our findings also highlight COTPA section 6b as the most important indicator for tobacco control among adolescents. In line with our findings, evidence largely supports the stricter implementation of section 6b of COTPA decreased tobacco use in adolescents[81] and vice versa (increased tobacco use by adolescents to proximity and density of retail outlets near schools).[82] The ToFEI guidelines emphasised the Self-Evaluation Scorecard for Tobacco-Free Educational Institutions and this is well aligned with our study findings on the regular evaluation of these guidelines.[13] With the widespread digital infrastructure in the country, compliance and monitoring of the indicators of ToFEI guidelines could benefit from the use of mobile application-based systems as suggested by stakeholders in our study. Similar application-based solutions have been implemented in several Indian states[83] and globally[84 85] and has been reported to improve awareness and surveillance in and around schools.

Our study interviewees emphasised that improved enforcement and implementation require collaborative multisectoral partnerships, which include schools, parents, officials at all levels (national, state and district), local representatives, civil society and government health and non-health departments. The multistakeholder partnerships have been observed to be beneficial in reducing tobacco use in many parts of the world.[17 86 87] For instance, in New Zealand almost all educational institutes that have declared themselves 100% tobacco-free reported collaboration with external providers.[88]

Despite the novel findings presented in the paper, a few study limitations are noted. We have captured the perspectives of a diverse range of stakeholders representing different levels of governance (at national, state, district and village level) other than adolescents, thus the beneficiary perspective (adolescents) on tobacco control policies is missing. Sampling was one of the limitations of our study, as the interviewees for the in-depth interviews were purposely selected. Nevertheless, we had a fairly balanced response in terms of stakeholders from different levels of governance and years of experience in tobacco control making our study findings transferable and significant. Additionally, the lack of comparison of study findings to a framework can be looked at as a limitation that can be explored in future research.

## Conclusion

This study provides recommendations for strengthening the enforcement of existing, and developing new tobacco control programmes and policies focusing on adolescents to advance India towards the ultimate goal of a tobacco-free future generation and replicate such models globally.

**Acknowledgements** The authors would like to acknowledge the contributions of Dr Nishigandha Joshi for transcribing the data from in-depth interviews. The authors thank all the interviewees who participated in this research from renowned tobacco control organisations in India. The paper benefitted significantly from their insights and cooperation.

**Contributors** MA, JB, SB, MB and MMK developed the research questions and study methodology. SB and MB developed an interview guide and MA, MMK and VGK provided inputs to the interview guide. The list of interviewees was developed by MA, SB and MMK. SB, MMK and RB coordinated and conducted the interviews. Data was coded and thematically analysed by SB, MB and NT. SB, MB and NT drafted the manuscript. MA, JB, MMK, VGK, RB and IB critically reviewed the manuscript. All authors are responsible for the overall content as guarantors.

**Funding** This research work is funded by the Medical Research Council of the UK under the Global Alliance for Chronic Lung diseases programme (MR/P008933/1) and was carried out as part of a collaborative project of the University of Nottingham, Manipal Academy of Higher Education, HRIDAY and the University of Bath.

**Competing interests** None declared.

**Patient and public involvement** Patients and/or the public were not involved in the design, or conduct, or reporting, or dissemination plans of this research.

**Patient consent for publication** Not applicable.

**Ethics approval** The study was approved by the Indian Ministry of Health and Family Welfare Screening Committee (HMSC, 2017-0460), the Institutional Ethics Committee at the Centre for Chronic Disease Control (CCDC_IEC_11_2018), Manipal Academy of Higher Education (MAHE EC/012/2017) and the University of Nottingham, School of Medicine Ethics Committee (OVS200317).

**Provenance and peer review** Not commissioned; externally peer reviewed.

**Data availability statement** Data are available upon reasonable request. Data are available on reasonable request. As per our institutional data sharing policy, prior approval is required from the Research Management Committee (RMC) and the principal investigator (PI) of the study. After Committee's approval of the request, de-identified data can only be shared upon request.

**ORCID iDs**
Shalini Bassi http://orcid.org/0000-0001-6348-3335
Monika Arora http://orcid.org/0000-0001-9987-3933
Veena G Kamath http://orcid.org/0000-0002-0853-095X
Manpreet Bains http://orcid.org/0000-0002-1990-5948

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
