## [Reviewer comments · BMJ Open]

ARTICLE DETAILS

TITLE (PROVISIONAL)	Programme and Policy Perspectives Towards a Tobacco-Free Generation in India: Findings from a Qualitative Study
AUTHORS	Bassi, Shalini; Arora, Monika; Thapliyal, Nishibha; Kulkarni, Muralidhar; Bhagawath, Rohith; Bogdanovica, Ilze; Kamath, Veena; Britton, John; Bains, Manpreet

VERSION 1 – REVIEW

REVIEWER	Thai, Phong K Queensland University of Technology
REVIEW RETURNED	19-Sep-2022

GENERAL COMMENTS	This review was conducted by Ms Giang Vu, a PhD candidate at the University of Queensland This is an interesting study addressing the important issue of having effective adolescent-specific tobacco control policy. The authors have adopted appropriate methods for conducting the investigation and analysing the information collected. However, I believe that the Introduction and Results sections need major revision for this paper to be of publication standard. In particular, I suggest the authors to restructure and rewrite the Introduction to highlight (1) the need for adolescent-specific tobacco control policy, and (2) the importance of having stakeholders' inputs in the process of developing and implementing policies. The current Introduction, in my opinion, suffered from having too much irrelevant data/ statements that the argument facilitating the research question is somewhat lost. I found the current Results section hard to follow and thus need restructuring/ rewriting. The authors may consider categorized findings by groups of stakeholders/ level of authority. The use of sub-headings would be appreciated. In addition, I feel that the part on 'barriers and facilitators' need to be more emphasized, as it is currently lacking in the results. I found the lack of discussion on the influence of tobacco users at home to the adolescent should be addressed because this could seriously reduce the effectiveness of any other policies/intervention The abstract may also benefit from a thorough revision (especially the Results part) including proofreading and checking for grammar errors.
--

REVIEWER	Saito, Junko National Cancer Center Research Institute
REVIEW RETURNED	12-Nov-2022

GENERAL COMMENTS	This study focused on the perspective of adolescent's tobacco control measures by community stakeholders. Targeting various stakeholders from the national to the local level are significant, but there are some points that need to be improved. Results 1. It is recommended that the characteristics of the participants be summarized in a table. It is difficult to tell at a glance who and how many people participated at each level. Also, regarding one of the limitations of this study, the limitation of generalizability, what specific stakeholders were not included? What biases are possible? 2. Regarding the barriers and facilitators, it is unclear how many and what factors were eventually identified. Also, it would be more comprehensive if some framework (e.g. the Consolidated Framework for Implementation Research (CFIR)) is used to summarize the results. Also, it seems that some of the results of the recommendation included results that could be identified as barriers and facilitators. For instance, "effective and efficient inter-sectoral collaborations" may be categorized as a facilitator of successful implementation, and "lack of awareness about the ongoing program among members of enforcement agencies" may be a barrier. I would recommend to review the interview results throughout again to extract the barriers and facilitators comprehensively and clearly. 3. A strength of this study is that interviews were conducted with stakeholders at various levels, from national to local. However, the levels have not been given much consideration in the analysis of the interview data. For instance, an analysis that compares the interview data by level to identify differences and similarities would lead to more specific recommendations. Discussion 4. The purpose of the study is to explore stakeholder perspectives, influencing factors (barriers and facilitators), and recommendations, but the discussion seems to focus only on recommendations. It is unclear how the perspectives and influencing factors are related to the recommendations. For example, in the discussion, measures to overcome or promote the identified barriers and facilitators may be discussed as recommendations.
--

VERSION 1 – AUTHOR RESPONSE

Reviewer: 1

Dr. Phong K Thai, Queensland University of Technology
 Comments to the Author:
 This review was conducted by Ms Giang Vu, a PhD candidate at the University of Queensland
 This is an interesting study addressing the important issue of having effective adolescent-specific tobacco control policy. The authors have adopted appropriate methods for conducting the investigation and analysing the information collected. However, I believe that the Introduction and Results sections need major revision for this paper to be of publication standard.

In particular, I suggest the authors to restructure and rewrite the Introduction to highlight (1) the need for adolescent-specific tobacco control policy, and (2) the importance of having stakeholders' inputs in the process of developing and implementing policies. The current Introduction, in my opinion, suffered from having too much irrelevant data/ statements that the argument facilitating the research question is somewhat lost.

Response: Thank you for your suggestion. We have reframed the introduction section, as suggested by you highlighting the need for an adolescent-specific tobacco control policy and the significance of having stakeholders' inputs in the process of developing and implementing tobacco control policies.

I found the current Results section hard to follow and thus need restructuring/ rewriting. The authors may consider categorized findings by groups of stakeholders/ level of authority. The use of sub-headings would be appreciated. In addition, I feel that the part on 'barriers and facilitators' need to be more emphasized, as it is currently lacking in the results. Response: Thank you for your suggestion. The results section has been restructured to fall under the following subheadings to improve comprehensiveness.

- I. Existing tobacco control policy and programme actions focusing on adolescents in India**
 1. Barriers to the successful implementation of tobacco control programs and policies
 2. Facilitators to the successful implementation of tobacco control programs and policies
- II. Recommendations for the successful implementation and strengthening adolescent-specific tobacco control policies and programmes to advance India's transition towards a tobacco-free generation**
 1. Awareness generation and capacity building of enforcement agencies
 2. Strengthen the enforcement and monitoring of existing tobacco control policies and programmes
 3. Development of new tobacco-free generation-centric policies
 4. Multi-sectoral and multi-stakeholder engagement
 5. Regular impact evaluation

Also, the part on barriers and facilitators has been elaborated on after digging at the data level.

I found the lack of discussion on the influence of tobacco users at home to the adolescent should be addressed because this could seriously reduce the effectiveness of any other policies/intervention. The abstract may also benefit from a thorough revision (especially the Results part) including proofreading and checking for grammar errors

Response: Thank you for suggesting, a valid one. We have included a discussion on environmental (home) tobacco influence, as suggested. We have also made changes to the abstract after checking grammatical errors.

Reviewer: 2

Dr. Junko Saito, The University of Tokyo
Comments to the Author:
This study focused on the perspective of adolescent's tobacco control measures by community stakeholders. Targeting various stakeholders from the national to the local level are significant, but there are some points that need to be improved.

Results

1. It is recommended that the characteristics of the participants be summarized in a table. It is difficult to tell at a glance who and how many people participated at each level. Also, regarding one of the limitations of this study, the limitation of generalizability, what specific stakeholders were not included? What biases are possible? Response: Thank you for the suggestion. To bring out clarity at a glance, on the number of participants, their geographical representation and their characteristics, Table 1 has been

included in the manuscript to summarize the characteristics of the study participants. Also, perspective of all stakeholders, other than the beneficiary (adolescents) has been included in the study. Although we used purposive sampling, nevertheless we had a fairly appropriate representation from national, state and district levels, along with years of experience and their type of organization etc, making our findings transferable and significant.

2. Regarding the barriers and facilitators, it is unclear how many and what factors were eventually identified. Also, it would be more comprehensive if some framework (e.g. the Consolidated Framework for Implementation Research (CFIR)) is used to summarize the results. Also, it seems that some of the results of the recommendation included results that could be identified as barriers and facilitators. For instance, “effective and efficient inter-sectoral collaborations” may be categorized as a facilitator of successful implementation, and “lack of awareness about the ongoing program among members of enforcement agencies” may be a barrier. I would recommend to review the interview results throughout again to extract the barriers and facilitators comprehensively and clearly.

Response: Thank you for suggesting. Summarizing our findings in line with a framework may be too difficult and hence mentioned it as one of the limitations of our study. The findings specific to barriers and facilitators has been elaborated on after digging at data level.

3. A strength of this study is that interviews were conducted with stakeholders at various levels, from national to local. However, the levels have not been given much consideration in the analysis of the interview data. For instance, an analysis that compares the interview data by level to identify differences and similarities would lead to more specific recommendations. Response: Thank you for suggesting and we understand you call for suggesting segregation of data by the ‘level of involvement’. However, we would like to state that our aim of the study was to explore perspectives and then provide recommendations holistically, and thus instead we found segregating the findings as following more appropriate.

I. Existing tobacco control policy and programme actions focusing on adolescents in India

1. Barriers to the successful implementation of tobacco control programs and policies
2. Facilitators to successful implementation of tobacco control programs and policies

II. Recommendations for the successful implementation and strengthening of adolescent-specific tobacco control policies and programmes to advance India’s transition towards a tobacco-free generation

1. Awareness generation and capacity building of enforcement agencies
2. Strengthen the enforcement and monitoring of existing tobacco control policies and programmes
3. Development of new tobacco-free generation-centric policies
4. Multi-sectoral and multi-stakeholder engagement
5. Regular impact evaluation

Discussion

4. The purpose of the study is to explore stakeholder perspectives, influencing factors (barriers and facilitators), and recommendations, but the discussion seems to focus only on recommendations. It is unclear how the perspectives and influencing factors are related to the recommendations. For example, in the discussion, measures to overcome or promote the identified barriers and facilitators may be discussed as recommendations.

Response: Thank you for suggestion. The recommendations are based on study findings that included information on existing tobacco control policies and programs, barriers and facilitators for their successful implementation. Most recommendations are ideas provided by the participants to overcome identified barriers and promote facilitators in their opinion. For example; lack of awareness among teachers and implementing agencies was reported as a challenge/barrier by many and suggested recommendations overcome this barrier were to conduct sensitization workshops for teachers and members of implementing agencies.

VERSION 2 – REVIEW

REVIEWER	Thai, Phong K Queensland University of Technology
REVIEW RETURNED	12-Jan-2023

GENERAL COMMENTS	I appreciate the authors' effort in addressing my concerns. I found that most of my comments have been sufficiently addressed however I believe that the paper can benefit from further review and edit of language and writing style, as there are still minor language/ writing issues e.g. repeating of phases in the abstract
---

REVIEWER	Saito, Junko National Cancer Center Research Institute
REVIEW RETURNED	24-Jan-2023

GENERAL COMMENTS	Thank you for your response and revisions. The results are much easier to read now as subheadings have been added. However, the results chapter in the abstract could be improved. As the purpose of the study is to explore stakeholder perspectives, influencing factors (barriers and facilitators), and recommendations, the barriers and facilitators identified should be clearly and concisely stated separately from the recommendations.
---

VERSION 2 – AUTHOR RESPONSE

Reviewer: 1

Dr. Phong K Thai , Queensland University of Technology

Comments to the Author:

I appreciate the authors' effort in addressing my concerns.

I found that most of my comments have been sufficiently addressed however I believe that the paper can benefit from further review and edit of language and writing style, as there are still minor language/ writing issues e.g. repeating of phases in the abstract

Response: Thank you for your suggestion. As recommended, we have worked on writing quality for the entire manuscript, especially the abstract, keeping in mind your suggestions.

Reviewer: 2

Dr. Junko Saito, National Cancer Center Research Institute

Comments to the Author:

Thank you for your response and revisions. The results are much easier to read now as subheadings have been added. However, the results chapter in the abstract could be improved. As the purpose of the study is to explore stakeholder perspectives, influencing factors (barriers and facilitators), and recommendations, the barriers and facilitators identified should be clearly and concisely stated separately from the recommendations.

Response: Thank you for your suggestions. We have rewritten the result section of the abstract as per your suggestions. The barriers, facilitators and suggested recommendations have been listed separately under the result section.